# Longitudinal qualitative study of living with neurogenic claudication

Frances Griffiths ![ORCID],[1] Cynthia Srikesavan ![ORCID],[2] Lesley Ward,[3] Graham Boniface,[2] Esther Williamson ![ORCID],[2,4] Sarah E Lamb[2,4]

[1]Warwick Medical School, University of Warwick, Coventry, UK
[2]Nuffield Department of Orthopaedics, Rheumatology, and Musculoskeletal Sciences, University of Oxford, Oxford, UK
[3]Department of Sport, Exercise & Rehabilitation, Northumbria University, Newcastle upon Tyne, UK
[4]College of Medicine and Health, University of Exeter, Exeter, UK

**Correspondence to**
Dr Esther Williamson;
esther.williamson@ndorms.ox.ac.uk

## ABSTRACT

**Objectives** Neurogenic claudication (NC) causes pain and reduced mobility, particularly in older people, and can negatively affect mental and social well-being, so limiting successful ageing. This qualitative study explored how people with NC changed over 12 months.

**Design** A longitudinal qualitative study using semi-structured interviews.

**Setting** Participants were recruited from a UK clinical trial of a physiotherapy intervention for NC.

**Participants** Interviews were undertaken at baseline, 1 month after receiving any intervention and at 12 months. We analysed 30 sets of three interviews.

**Results** Interview data were summarised for each time point into biopsychosocial domains: pain, mobility and activities of daily living, psychological impact, and social and recreational participation. Through comparative analysis we explored participant trajectories over time. Progressive improvement in at least one domain was experienced by 13 participants, but there was variability in trajectories with early improvements that remained the same, transient changes and no change also commonly observed.

Eleven participants described co-present improvement trajectories in all domains. Three participants described co-present improvement in all domains except participation; one had never stopped their participation and two had unattainable expectations. Five participants described co-present improvement in one domain and deterioration in another and 14 participants described co-present no change in one domain and change in another. There was evidence of interaction between domains; for example, improved mobility led to improved participation and for some participants, specific factors influenced change. Of the 15 participants who experienced improved participation, 10 reported improvements in all other domains and five participants did not; for two, pain did not prevent participation, one used a walking aid and two had a positive psychological outlook.

**Conclusion** The daily lived experiences of older adults with NC are variable and include interaction between biopsychosocial domains. Therapist understanding of these trajectories and their interactions may help to provide personalised therapy

**Trial Registration Number** ISRCTN12698674

## STRENGTHS AND LIMITATIONS OF THIS STUDY

⇒ In-depth interviews at three time points focus on participant's current experience; comparison across time was undertaken during analysis so was not reliant on participant interpretation of past experience.
⇒ Analysis summarises succinctly a large qualitative data set.
⇒ Summarising data will have resulted in loss of nuance in our analysis.
⇒ The results present one set of trajectory category membership for participants; some participants could be categorised differently.
⇒ Our interest in successful ageing shaped our path through analysis; a different interest is likely to result in different trajectory descriptions.

## BACKGROUND

Neurogenic claudication (NC) is caused by narrowing of the spinal canal (lumbar canal stenosis) most often due to osteoarthritis of the spine or spondylosis. It is frequently reported in adults aged above 60 years.[1 2] NC usually presents as pain and discomfort radiating from the back into the buttocks and legs in a unilateral or bilateral pattern. It is provoked by walking or prolonged standing and relieved by sitting or lumbar flexion.[3] The pain often occurs intermittently. Other signs and symptoms may include weakness, altered sensations such as pins and needles and numbness, fatigue and gait changes.[3] Pain in the lower back is usual but not a necessary diagnostic feature but regardless NC is considered a low back pain condition. NC is associated with decreased mobility and independence for older people, restriction of social activities, a negative emotional impact and frailty.[4 5] The impact of back and leg pain consistent with NC on health status is more marked than for people with other patterns of back and leg pain; for older adults, compared with those who report back pain only or have leg pain which is not consistent with NC, those with NC report lower quality of life and are more likely to be frail, be less confident to walk half a mile and to report a decline in their mobility over the last year.[1] The variation in symptom trajectory for people with NC is illustrated by a study of 146 people with NC followed up for 3 years (without surgical

treatment); symptoms improved for about one-third of patients, remained unchanged for half the patients and worsened in a few.[6]

Ageing successfully is considered a key mechanism by which ageing populations can maximise their health and longevity and minimise their need to access healthcare systems.[7] The concept of successful ageing considers physical and mental health along with social and recreational engagement[7] and since its conception has been expanded to include living with multi-morbidities and chronic conditions such as NC.[8–10] Our concern in this study is whether the symptoms of NC may lead to older people being unable to age successfully. We were particularly concerned to understand whether and how NC impacted on social and recreational participation and how this changed over time.

### Qualitative evidence on living with back pain

The interaction physical and mental health and social and recreational engagement has been explored in qualitative studies of people living with low back pain and synthesised in a meta-ethnography.[11] The inter-related themes identified were impact of pain on self, use of coping strategies and impact of pain on engagement with others. The meta-ethnography concluded that chronic low back pain is complex, dynamic and multidimensional with those living with the pain experiencing distressing pain and loss with lowered self-worth, stigma, depression, premature ageing and fear of the future. A later cross-sectional qualitative study of people with NC suggests that pain, physical ability and emotions are inter-related and at times inseparable.[5] The meta-ethnography identified a lack of longitudinal studies. A subsequent longitudinal study interviewed eight people living with back pain annually over 2 years.[12] Although pain and loss impacted on all areas of the participants' lives, two different patterns of change over time were identified: participants who saw their pain as a physical problem and a cause of loss and did not re-establish social and recreational activities, and those who experienced pain relief and were less enmeshed in the pain experience so able to look to the future.[12]

Such longitudinal qualitative studies provide a depth of understanding to complement existing panel studies[13] and build on the idea of illness trajectory,[14] biographical disruption[15] and understanding individuals as unique, complex and adaptive.[16–18] Our study aims to provide this depth of understanding of people living with NC: how their NC impacts on their function and engagement and how this changes over time. It builds further on longitudinal qualitative studies in the health domain.[19]

### BOOST trial

This qualitative study was embedded within the Better Outcomes for Older people with Spinal Trouble (BOOST) trial,[20 21] a multi-centre, randomised controlled trial with two treatment arms. It compared a 12-week group-based physical and psychological intervention (the BOOST programme) to individual best practice advice (BPA) for older adults (65+ years) with NC. The BOOST programme comprised group discussions that were underpinned by cognitive–behavioural techniques, an individually tailored exercise and walking circuit, and home exercises. BPA comprised home exercises and self-management (one to three appointments with a physiotherapist). Participants aged 65 years or over with the clinical presentation of NC symptoms were recruited from primary care practices of the National Health Service (NHS) in the UK. People living in residential care or nursing homes, those registered blind and those with a terminal health condition or on a waiting list for surgery or unable to walk 3 m without the support of another person or unable to provide informed signed consent were not included. Participants were randomised in a 2:1 ratio (BOOST programme: BPA). Full trial protocol details are reported elsewhere.[21]

## METHODS

### Study design

We used a longitudinal qualitative research design to understand the individual at the time of each interview[22] rather than seeking from the individuals their own narrative of change.[23] We aimed to interview participants three times: at trial baseline (after randomisation but before the participant knew their treatment allocation) (T1), 1 month after any intervention (T2) and 12 months post-randomisation (T3). T1 interviews were via face-to-face at participant's home, with T2 and T3 by telephone. Where participants had difficulty using the telephone, T2 and T3 interviews were undertaken face-to-face.

### Participants and recruitment

All participants recruited to the BOOST trial[20 21] were eligible for the interview study and consent to invite participants for interview was collected as part of trial procedures. We invited participants consecutively from each recruitment site over the 25-month trial recruitment period (August 2016 to August 2018), stratifying to ensure balance between range of age, gender, ethnicity, treatment allocation (ratio 1:1) and recruitment site location. To ensure sufficient information power,[24] we aimed for 30 participants retained at 12 months and so aimed to recruit approximately 60 participants to allow for 50% attrition.

### Data collection

We obtained written consent at the start of the first interview and reaffirmed this verbally at follow-up interviews. We used semi-structured interview schedules (online supplemental file 1). We used the present tense when asking questions to encourage participants to focus on their current state.[22] In T1 interviews, we explored the impact of NC on participant physical and psychosocial health including NC and their beliefs around exercise and ageing. In T2 and T3 interviews, we explored similar issues along with their experiences of the BOOST trial

interventions. Before T2 and T3 interviews, we reviewed for each participant the data captured in their earlier interviews and made notes about the participant and their context. We used this knowledge in the interviews to rapidly establish rapport on the telephone and enable quick clarifications. For example:

Participant: I go to my club.

Interviewer: Is that the ladies club you mentioned last time we met?

All interviews were audio-recorded onto encrypted recorders and stored securely at University of Oxford. During transcription/summarising, all personal identifiers were removed.

## Patient and public involvement (PPI)
A group of patient representatives were involved in developing the application for the BOOST study and one member was co-applicant. PPI representatives reviewed patient materials such as consent forms, participant information sheet and the interview schedules. We carried out some practice interviews with PPIs to test and refine the interview schedules prior to data collection. The interview findings were presented to the PPI representatives in a study meeting and were reviewed by the lead PPI representative.

## Data analysis
Throughout our analysis, we compared data across time and between participants and drew on existing evidence-based concepts and definitions related to back pain and NC. We mapped the data into a framework matrix[25] then explored the matrix[26 27] for change processes related to NC. This involved summarising patterns of change being experienced at each time point, trajectories of change between time points,[27] co-presence of trajectories, and interactions between trajectories and turning points. In our analysis of data at each time point, we used an analysis approach we had previously developed[18] and applied.[22] We focused on current experience and put to one side data where the participant was interpreting past experience. This allowed us to compare what was current at each time point and to manage the volume of complex data.[28] We describe each analysis step. Analysis was conducted in Excel.

### Step 1: development of individual participant frameworks
We started analysis on complete interview sets (T1, T2 and T3). We selected sets where the participant provided rich data about the experience of NC. After selecting 30 sets (90 interviews) and undertaking initial analysis (see step 2), we read and discussed our remaining data to check whether these were revealing new themes; we determined we had achieved data saturation.[29] All T1 interviews were transcribed verbatim along with the first 10 T2 and 10 T3 interviews we analysed. The remaining interviews were analysed directly from audio recordings. To immerse ourselves in this large volume of data, we initially wrote structured pen portraits[30] at each time point for 10 data sets. This involved reading and rereading the data, clarifying our focus on current experience related to NC, summarising participant experiences and context with illustrative quotes, and developing a structure that worked across the 10 data sets. From these, we developed and refined a coding template. We used this to thematically code all 30 interview sets, writing mini-statements with illustrative quotes for each relevant code/subcode at each time point. The resulting 30 individual participant frameworks were used for subsequent analysis.

### Step 2: development of change framework
After team discussions, we summarised the coded data for each participant at each time point into the following biopsychosocial domains: pain, mobility and activities of daily living (ADL), psychological impact, social and recreational participation, and sleep. For example, to summarise mobility status at T1, we collated and summarised data from codes (current mobility, activities reduced, activities stopped, activities increased) and subcodes such as physical impact. Participants reported a range of symptoms including numbness, tingling and heaviness. However, the symptom that consistently predominated across all participants' accounts of their experience of NC was pain; we therefore summarised the experience of pain in the framework. We included data that conveyed the pattern of change each participant was experiencing at the time of each interview, for example, a gradual improvement or worsening as experienced day-to-day/week-to-week. We added to the change framework participant age, gender, ethnicity, living arrangement, group allocation in the trial, health events and co-morbid conditions. We summarised participant coping strategies, views on ageing, use of mobility aids and feedback from others.

### Step 3: identification and categorisation of participant trajectories
To identify participant trajectories for each biopsychosocial domain, we compared the summaries in the change framework for what increased or emerged over time, what decreased or ceased over time, and what remained consistent over time.[26] We added a description of this trajectory to the change framework. In a new table, we presented for each domain the colour-coded trajectory for each participant. Each team member then independently, by eye, ordered the participants so those with the most similar combination of trajectories were adjacent. The team then compared their resulting tables, discussed similarities and differences, and agreed on one version of the table. Using this table, we then independently categorised the participants into groups based on similarity of trajectories and discussed and agreed categorisation.

### Step 4: exploring the nature of the trajectories
*Identifying and categorising co-presence and interaction between the trajectories of the individual biopsychosocial domains*
We scrutinised the change framework for each participant and compared participants to understand which trajectories were co-present at each time point and so may have interacted with each other. For example, a reduction in pain was accompanied by improvement in mobility or deterioration

in psychological outlook was accompanied by reduced social or recreational participation. We returned to participant data to check whether the participant reported interaction between the trajectories. We grouped similar types of co-presence and explored how these related to our trajectory categories. Given our interest in successful ageing, we particularly explored how social and recreational engagement changed along with other domains.

### Identification of health-related turning points

We examined the data for health-related turning points mentioned by the participant; what the participants considered significant change in their health during the study timeframe.

### Totality of change

From the whole change framework, we examined factors that were not inseparable from trajectory category although are likely to be inter-related.[5] Factors included whether they lived alone, how they considered ageing and use of mobility aids, and strategies for coping with NC in their daily lives. We identified what was similar and different between participants in the different trajectory categories.

### Study team roles

SEL and EW (health service researchers and physiotherapists) led the BOOST programme of work including this study and the BOOST trial. FG (general medical practitioner, medical sociologist and qualitative researcher) led the qualitative study team that included LW (qualitative health researcher), CS (physiotherapist and a qualitative researcher) and GB (physiotherapy researcher). All contributed to qualitative study design, reading data, thematic and comparative analysis, synthesis, interpretation and reporting. LW undertook all interviews. LW and FG reviewed the interview schedules to improve the clarity and flow of the questions. CS worked with FG on analysis with GB providing independent review of coding/ summaries. Any discrepancies were resolved through discussion. EW and SEL reviewed each step of the analysis and provided feedback to FG, LW, CS and GB.

This study team were not involved in the delivery of BOOST interventions or outcome assessments in the trial. FG, LW, CS and GB remained blinded to the results of the BOOST trial until they had completed data coding and analysis.

### Reporting

The Standards for Reporting Qualitative Research guidelines[31] (SRQR) were used to report the study. Quotations from interviews are labelled with participant number and interview T1, T2 or T3.

## RESULTS
### Data

Participant attrition across the three time points was lower than expected, with 49 of the 60 participants (82%) providing three interviews. Reasons for withdrawal were as follows: withdrawal from clinical trial (n=3), ill health/awaiting investigations (n=4), increased pain after receiving BOOST programme sessions (n=2), declined third interview (n=3), unable to contact (n=3) and withdrawn by researcher as participant postponed >6 months (n=2). Initial interviews averaged 70 min (range 44–93 min), with second and third interviews averaging 36 min and 32 min, respectively.

### Participant characteristics

There was little difference in terms of age (average 75 years), gender and ethnicity between trial, interview and analysis cohorts, and were similar in terms of measures related to NC (table 1).

### Analysis results
#### Analysis steps 1 to 3: participant trajectories

From summarising patterns of change from our individual participant frameworks (example available on request) into our change framework (online supplemental file 2), we identified patterns of change for each participant in each domain (online supplemental file 3). We then summarised the trajectory across the three time points (see table 2). We omitted sleep as a domain as although a question on sleep was asked in each interview, many interviewees did not provide data.

Progressive improvement in at least one domain was experienced by a total of 13 participants. Trajectories of no change, early improvements that remained the same and transient changes predominated. Of the 30 participants, 16 experienced no improvement in at least one domain. Improvement in pain was experienced by 16 participants. All 15 participants who experienced improved social and recreational participation also reported improvements in other domains (although for one the improvement in another domain was after initial worsening).

The trajectories in any one domain are not an indication of the amount or nature of change so the same trajectory might be experienced very differently. For example, participant 24 and participant 50 both showed progressive improvement in mobility, but one was able to walk much further than the other:

> And as I said, I can go out and I can go off to different parts of the shops, or garden centres, we like looking round those (24, T3).

> I am able to get up and walk without any aids around it [home] now (50, T3).

In the domain of psychological impact, participant 04 and 02 continued from T1 to T3 with negative but different experiences. The first experienced frustration as he had to rely on his wife to do his self-care activities:

> I can't get myself dressed, I can't do the things that I would normally do or anybody else would normally

**Table 1** Description of BOOST trial participants, those interviewed and those included in the qualitative analysis presented in this paper

| Participant characteristics | BOOST trial cohort N (%) Mean (SD) Median (IQR) | Interview cohort N (%) Mean (SD) Median (IQR) | Interview analysis cohort N (%) Mean (SD) Median (IQR) |
|---|---|---|---|
| Total participants | 435 | 59 | 30 |
| Age (years at baseline) | 74.9 (6.0) | 75.5 (6.2) | 75.1 (5.7) |
| Gender | | | |
| Female | 246 (56.6) | 31 (52.5) | 14 (46.7) |
| Male | 189 (43.4) | 28 (47.5) | 16 (53.3) |
| Ethnicity | | | |
| White British/White Other | 400 (92) | 52 (87) | 26 (86.6) |
| Black British | 12 (2.8) | 3 (5) | 2 (6.7) |
| Indian | 9 (2.1) | 3 (5) | 2 (6.7) |
| Pakistani | 6 (1.4) | 1 (2) | 0 (0) |
| Confidence walking long distance | 5.6 (3.3) | 5.1 (3.3) | 4.7 (3.0) |
| Oswestry disability index | 32.9 (13.9) | 32.7 (13.5) | 34.6 (14.5) |
| Tilburg frailty index | 4.5 (2.6) | 4.8 (2.8) | 5.6 (2.8) |
| Exercise self-efficacy, median (IQR) | 69 (53, 80) | 69 (43, 86) | 52.5 (37, 81) |
| Fear avoidance beliefs | 12.9 (5.9) | 14.0 (6.3) | 14.1 (6.4) |
| Six-minute walk test | 255.4 (99.1) | 249.3 (96.9) | 227.7 (91.9) |
| Self-management rating scale | 6.08 (1.80) | 6.0 (1.7) | 6.1 (1.7) |
| **Treatment allocation in the trial*** | | | |
| BOOST programme (BOOST) | 292 (67.1) | 32 (53) | 16 (53.3) |
| Best practice advice (BPA) | 143 (32.9) | 27 (45) | 14 (46.7) |
| **Complete sets of three interviews** | | 49 | 30 |

Scale range and direction:
Confidence in ability to walk long distance (1 to 10, lower values worse)
Oswestry Disability Index V.2.1a (0 to 100, higher values worse)
Tilburg Frailty Index (0 to 15, higher values worse)
Exercise self-efficacy scale (0 to 90, lower values worse)
Fear Avoidance Beliefs Questionnaire Physical Activity (0 to 24, higher values worse)
Six-minute walk test (distance covered walking for 6 min, lesser distance worse)
Self-management rating scale (0 to 10, lower values worse)
Values are N (%) or mean (SD) or median (IQR) as specified.
BOOST, Better Outcomes for Older people with Spinal Trouble.

do for themselves. I need a hand; I need help with that, which I do get obviously, with [wife]. Again, that's frustrating because she's got things to do and I don't feel I can keep saying to her, oh can you just help me with this, can you help me with that? (04, T3)

The second describes fear of damaging his back with certain activities:

But I don't do gardening like I used to do before. For the fear of damaging or hurting my back, and things like that. Because I tried that in the past, and once I start doing something then I get carried away, and the next two days then I have to suffer, so I try and avoid it (02, T3).

In the domain of social and recreational participation, participants 07 and 24 experienced progressive improvements over time but with different reach. The first was prepared to extend her activities:

But wherever I go, if I go on holiday as well, I never say no, that I can't do (some)thing (07, T3).

The second was positive about her ability to undertake small tasks that helped her feel she was participating equally with others. She described social outings with her friends:

I can carry a tray with drinks on now. I can take my turn, I can get in the queue, where it always used to be 'you get the table, I'll go' (24, T3).

**Table 2** Participant trajectories in the biopsychosocial domains

| Categories of trajectories | Participant ID | Pain | Mobility & ADL | Psychological impact | Social and recreational participation |
|---|---|---|---|---|---|
| **Category 1 (n=10)** Participants with progressive improvement in at least two domains | 08 | | | | |
| | 47 | | | | |
| | 30 | | | | |
| | 57 | | | | |
| | 50 | | | | |
| | 25 | | | | |
| | 24 | | | | |
| | 07 | | | | |
| | 01 | | | | |
| | 05 | | | | |
| **Category 2 (n=5)** Participants with initial and maintained or delayed improvement in at least two domains | 37 | | | | |
| | 43 | | | | |
| | 26 | | | | |
| | 31 | | | | |
| | 09 | | | | |
| **Category 3 (n=5)** Participants with discordant trajectories – where at least one domain trajectory was to some extent in the opposite direction to at least one other domain trajectory | 38 | | | | |
| | 15 | | | | |
| | 02 | | | | |
| | 04 | | | | |
| | 41 | | | | |
| **Category 4 (n=3)** Participants with transient change in at least two domains | 39 | | | | |
| | 14 | | | | |
| | 33 | | | | |
| **Category 5 (n=7)** Participants with no change in at least two domains | 34 | | | | |
| | 32 | | | | |
| | 53 | | | | |
| | 06 | | | | |
| | 28 | | | | |
| | 03 | | | | |
| | 11 | | | | |

**Legends for trajectories**

- No change: No change in pain or in the decreased mobility decreased social and recreational participation or negative psychological impact experienced since before the onset of NC.
- Progressive worsening: Progressive worsening of pain, decreased mobility, decreased social and recreational participation, or negative psychological impact (i.e. continued worsening after an earlier worsening).
- Improvements remained same: Improvements experienced at T1 (compared to entry into trial) or at T2 (compared to T1) in pain, mobility, social and recreational participation, or psychological impact were maintained.
- Progressive improvements: Progressive improvements in pain, mobility, social and recreational participation, or psychological impact (i.e. continued gain on an earlier improvement).
- Transient change pattern 1: Initial improvements between T1 & T2 then reversed to T1 (Baseline) status at T3.
- Transient change pattern 2: Initial improvement between T1 & T2 & then worsening between T2 & T3, to worse than T1
- Transient change pattern 3: Initial worsening between T1 & T2 & then improving between T2 & T3
- Delayed improvement: Pain, mobility, social and recreational participation, or psychological impact improved at T3 only.
- Lack of sufficient information (at least 2 time points) to describe the change process.

ADL, Activities of daily living.

When placing participants adjacent to other participants based on the similarity of trajectory, we were making choices about which trajectories to prioritise. Different choices resulted in a different ordering of participants. For example, we could place together participants with two or three domains with the same trajectory and not worry about the trajectory of the other domain(s), or we could place together participants with the same number of trajectories of a certain type, irrespective of which domains they were. However, participants tended to remain towards one end or the other of the table. Table 2 presents one way of ordering the participants on which the team could agree.

Our visual impression was there were five main patterns of trajectories which we identified as our trajectory categories, although the boundaries of these are not clear cut. They were as follows: category 1 (n=10) participants with progressive improvements in at least two domains; category 2 (n=5) participants with initial and maintained or delayed improvements in at least two domains; category 3 (n=5) participants with discordant trajectories—where at least one trajectory was to some extent in the opposite direction to at least one other trajectory (the opposite direction might be transient); category 4 (n=3) participants with transient change (in the same direction) in at least two domains; category 5 (n=7) participants with no change in at least two domains. We illustrate the trajectories for one participant at each time point with quotations from their data in table 3 and for one participant from each trajectory at T3 in table 4.

This categorisation was developed from one way of ordering the participants (see above). It was therefore expected that some participants could be in more than one category (for example participant 34 could be in category 2 or 5).

We chose to define category 3, discordant trajectories, as trajectories going in opposite directions (improving/worsening). Arguably, a no change trajectory alongside a transient change or an improving/worsening trajectory is also discordant. All but one of the 10 participants in categories 4 and 5 have this latter combination of trajectories, and 4 of the 15 participants in categories 1 and 2.

Of the 15 participants who experienced improvement in social and recreational participation, 10 experienced improvements in all other domains and were in categories 1 and 2. The remaining five participants (24, 43, 38, 41 and 34) did not experience improvement in all other domains and were from across all categories except category 4.

**Table 3** Illustrative quotes for one participant (category 3) for all domains and time points

| ID: 04 | Pain | Mobility and ADL | Psychological impact | Social and recreational participation |
|---|---|---|---|---|
| T1 | 'And then, umm, there's pain if I get up, there's pain if I walk, there's pain sitting down, there's pain standing up, there's pain reaching out… it's a constant pain. 24/7 I'm in pain' | 'I'm inactive; I can't do anything without being in pain…' | '"So I started to get depressed then, because I couldn't… I can't do anything and I couldn't do anything and I, I just sort of thought, 'Well what's the point, what is the point of me being here when I can't do anything?' This is it, I'm just sitting around, I'm doing nothing, I'm no good …'' | 'Oh I love to see the grandkids but, again, I can't do anything with them, I can't do nothing with the grandkids because, again, the… My grandson's into football now and I feel there's a lot I could teach him with the, with the game and I can't.' |
| T2 | 'Not really, it's (Pain) there the majority of the time.' | 'Well every day, every day. I mean we're either out shopping or that, I don't think there's a day goes by where I'm not doing some sort of walking.' | 'I'm in a situation where I'm still depressed over it because obviously I'm not … I didn't expect a lot from it, I suppose in my mind I thought it might be, I might be having a lot more movements and things like that, which haven't worked out.' | 'Socialising I don't do. I'm not a socialiser anyway. All them days … since the football finished and everything I'm not, I'm not one for going out and that anyway. As I said, sometimes it's not worth the effort to get ready and. Getting washed and changed and then getting out like that, so much is involved with it, sometimes it's not worth doing it' |
| T3 | 'Well I'm still in a lot of pain….' | 'I feel more confident when I'm in the house. I meanwhile the weather's like this I won't budge out at all, I won't venture out at all, there's no point…. Obviously, when it's really severe, I can't get myself dressed, I can't do the things that I would normally do or anybody else would normally do for themselves' | 'I mean, I keep bringing the mental side of it because I think everything that goes on does affect you mentally. It's very hard not to get away from a lot of things mentally' | 'I said, my one grandchild is playing football himself now and I can't, there's no way I can go with him and help him or be involved in his, in coaching with him or teaching him anything or just having a kick around with him. I can't do that, so I feel I'm missing out on all that.' |
| Trajectory | No change | Initial improvements between T1 & T2 then reversed to T1 (Baseline) status at T3 | No change | Progressive worsening |

**Table 4** Illustrative quotes of participants from each trajectory category at T3

| Participants from trajectory categories | Pain | Mobility and ADL | Psychological impact | Social and recreational participation |
|---|---|---|---|---|
| Category 1 ID: 08 | 'But in terms of my back pain and legs, although I still have a little and last night it was particularly uncomfortable, most of the time it actually is fine, and I feel a lot better than I did last year.' | 'If I go for a walk, for example, with my wife, I can probably do quite a long-ish walk, probably three to five miles, which I couldn't do 12 months ago.' | 'But, for example, to give you a clue of how more confident I am, we are going away for 5 days on Sunday to (city) to do some sightseeing, some walking, and some photography. So I feel confident enough to be able to go and do that.' | '… we went down to (Town) for a week (Holidays), back in November and the apartment we had was about a mile and half out of the town centre. So we walked that both ways, in and out, most days, and I coped with that quite well.' |
| Category 2 ID: 31 | 'Oh it is better. It's under control, as much as I can get it under control, with the help of the Gabapentin, which, I know it's another drug to be on but if it works then.' | 'I walk down the stairs and up the stairs, rather than use the (stair lift). And I walk up and down the hall quite a bit, I do an exercise where I just walk up, turn around and come back, I do that about ten times. Particularly if it's a wet day and I can't get out.' | 'Well, I'm walking better. Whereas before I was getting frightened to walk, because I was in so much pain.' | 'Well, I tend to go in (Gardening) for a short while. Once my back starts hurting, I come back in and sit down. I don't overdo it generally.' |
| Category 3 ID: 04 | 'Well I'm still in a lot of pain, but not as much as I was last week. It's sort of calmed down a little bit now.' | 'Well, I don't cook for myself or anything like that. I mean, I can't mess around with things like that. Obviously, when it's really severe, I can't get myself dressed, I can't do the things that I would normally do or anybody else would normally do for themselves.' | 'I mean, I keep bringing the mental side of it because I think everything that goes on does affect you mentally. It's very hard not to get away from a lot of things mentally.' | 'Normally I miss out on a lot of things because of sitting somewhere for too long or something because this again causes discomfort. So rather than sit there feeling miserable I'd rather not go.' |
| Category 4 ID: 14 | 'At the moment I think I'm at a bad stage, because I've got so many things going on … I've got the back pain. I've got the neck pain. I've got the knee, which comes out of joint. So it's not a pretty sight really.' | 'But I still make myself, when we've got a good day, not today like it is now, but if it's a good sunny day I will still try to go out and do about ten minutes, fifteen minutes' walk.' | 'I was doing very well and very eager to get better or help myself somewhat … So I've got all these things (other musculoskeletal issues) going at once. Which has dragged me down a bit, there's no doubt about it, it's made me quite depressed.' | 'It's obviously; it's something I'm not used to. I'm used to going everywhere and very, sort of, doing exercises and yoga classes and everything else. And it's getting worse, because it's stopped, it's stopping, everything is going wrong.' |
| Category 5 ID: 11 | 'Once I've got my legs, once I'm moving a bit, it does seem to ease off, I'm not so bad. But if I sit down, then once I get up again it's back to square one; you're still in pain and tingling.' | 'So it's hard for me to put, you know, for me to actually sit still all day. In the last week, I've watched more telly than I've watched in 12 months. So you can tell, like, the situation. I've never watched so much, I've got square eyes.' | 'Well at the moment I'm more anxious than ever with what I've got now. So I'm still a bit anxious, well even before this, before this I was still anxious. I was quite anxious really, you know, because I wanted to do things.' | 'And so it's really, my lifestyle really, to me, it's gone to nil from what I - I'd be out doing everything sort of thing, nine o'clock, ten o'clock at night time gone by, til it got dark, pottering about, cutting the lawn, doing the lawn, or doing the neighbour's if they couldn't do it, elderly people I'd cut theirs for them. So it's, to say, like, you've come to, not nil as such, but you've really, it's a total cut-off …' |

### Analysis step 4: nature of the trajectories
*Co-Presence of domain trajectories and perceived interaction*

We identified 10 types of co-presence of the trajectories (table 5). The most common presentation was type 1 where positive trajectories were co-present for all domains. Our data suggest that improvement in one domain often led to improvement in another and so on (see illustrative quotes table 5). Type 2 (n=3) was similar except those improvements in pain, mobility and psychological were not co-present with improvements in social and recreational participation. The reasons for this were very different and specific to each individual: participant 09 was a 79-year-old farmer who was unable to get back to his heavy farm work, participant 37 was an 88-year-old who did not get back to going on group trips which involved walking as he did not want to hold up the others and 01 was an 81-year-old who at baseline had already managed to return to tai chi and her voluntary work and maintained this. All the participants of types 1 and 2 were in trajectory groups 1 and 2 (improvement trajectories)

**Table 5** Co-Presence of trajectories (⇑ improving; ⇓ worsening; ⇔ no change) and illustrations of how they interact (or not)

| Co-Present trajectories of biopsychosocial domains | Participant ID | Participants' quotes illustrating interaction between co-present trajectories | Trajectory category (see table 2) |
|---|---|---|---|
| (1) ⇑Pain, Mobility, Psychological impact, Participation | 30, 31, 07,08, 25, 38, 05, 26, 57, 47, 50 | 'When you first saw me, I was in a bit of pain, and everything was a struggle. But now, I take everything in my stride now… I think at least 75% I feel better in myself [Pain]… Oh I'm quite satisfied [with walking] at the moment. As I say, a couple of miles is easy now, when I go out shopping or anything like that [Mobility and ADL]… And then as I say, Wednesdays and Thursdays I go out to the club, have a drink, and socialise with my friends. And Saturday night I go out, get on the dance floor if I feel up to it, for half an hour or something like that [Participation]… But as far as, it [Neurogenic claudication] doesn't stop me doing anything I want to do' [Psychological impact] (30, T3) | 1 and 2 Improving |
| (2) ⇑Pain, Mobility, Psychological impact ⇔ Participation | 01, 09, 37 | 'I really haven't had any problems with my back (37, T2) [Pain]… I don't think there is any problem with my general health… I think I am walking better than I was when I started this scheme… I suppose I have really not a lot of problem, I can walk the distances that I need to now, I don't want big distance… I think it [Walking uphill] is easier… it has been easier before when I first met you, if I walked uphill, I felt exhausted…now I don't [Mobility and ADL]… I design my life around my abilities [Psychological impact]… Not a lot, to be fair (impact of condition on lifestyle)… I keep myself involved that is my objective to keep myself occupied… To be fair, I think I work my physical life to the limitations I have got and being my age… I don't suppose I can say it [General lifestyle] altered other than doing exercises in the morning' [Participation] (37, T3) | 1 and 2 Improving |
| (3) ⇔ Pain ⇑ Mobility, Psychological impact, Participation | 43, 24 | 'No, just through the night [Pain]… I'm getting around fine… I just feel good [Mobility and ADL]. Makes me feel good, it gives me confidence. You see I can do it, and now I know I can do it [Psychological impact]… I can carry a tray with drinks on now. I can take my turn, I can get in the queue, where it always used to be 'you get the table, I'll go" [Participation] (24, T3) | 1 and 2 Improving |
| (4) ⇓ Pain, Mobility, Psychological impact, Participation (except 41 where improved Participation gained by remained the same) | 03, 11, 33, 41, 28 | '… once I've got my legs, once I'm moving a bit, it does seem to ease off, I'm not so bad. But if I sit down, then once I get up again it's back to square one; you're still in pain and tingling [Pain].… I'm really basically, not doing nil, but you know, walking up and down the garden and walking, not really doing to what I normally do. It's really halved [Mobility and ADL]… Well at the moment I'm more anxious than ever with what I've got now. So I'm still a bit anxious, well even before this, before this I was still anxious. I was quite anxious really, you know, because I wanted to do things [Psychological impact]… And so it's really, my lifestyle really, to me, it's gone to nil from what I—I'd be out doing everything sort of thing, nine o'clock, ten o'clock at night time gone by, till it got dark, pottering about, cutting the lawn, doing the lawn, or doing the neighbour's if they couldn't do it, elderly people I'd cut theirs for them. So it's, to say, like, you've come to, not nil as such, but you've really, it's a total cut off' [Participation] (11, T3) | 5 No change |
| (5) ⇔ Pain ⇓ Mobility, Participation (or not mentioned). ⇑ Psychological impact | 53, 06 | 'Still I am in some difficulty and problem [Pain]… but… honestly I just carry on with my life [Psychological impact]…. It [Pain] is not too bad at the moment.… To be honest, I don't do anything [Mobility and ADL] at all… Some days… bad days which you know you just wonder what else you can do… Housework otherwise, I take my time. I don't try to push myself to you know everything I need to suppose one time… Little at a time [Mobility and ADL]… I don't worry about it I just try on and carry on with my life [Psychological impact]…. I do that [Shopping] not alone but always have somebody with me' [Participation] (06, T3) | 5 No change |

Continued

**Table 5** Continued

| Co-Present trajectories of biopsychosocial domains | Participant ID | Participants' quotes illustrating interaction between co-present trajectories | Trajectory category (see table 2) |
|---|---|---|---|
| (6) ⇔ Pain ⇓ Mobility ⇑ Psychological impact, Participation | 34 | 'Could [Pain] be better. I can feel my sciatic nerve all the time [Pain]… but it didn't really bother me, you know [Psychological impact]…. I can't walk as quick as I used to and I can't do steps… I am walking normally but the smaller steps what I used to do…. My legs hurt when I start moving around…. there are certain movements I just be careful with… but other than that, fine [Mobility and ADL], I am doing my garden. My back doesn't like it very often… You know I come in, sit down, have a cup of tea and go back again. I go to my gardening club… Tuesday I see my friend as usual… put the world right and have a cup of coffee… today I had been to school… so listen to the children reading and now being given an award today [Participation]…. You know you just have to get on with it, don't you? I can't go anywhere, I feel shattered, but you still have to do don't you? If you got to do something, you got to do it' [Psychological impact] (34, T3) | 5 No change |
| (7) ⇑ Pain ⇓ Mobility, Psychological impact ⇔ Participation | 02 | 'But my back pain, the constant pain and the severe pain that I used to have is no longer there at the moment, so I'm happy to say that. However, it is not completely gone, and I get days, every now and again, when the pain comes back [Pain]…. What happens is then the pain starts, not while I'm walking but after I have done my walk and I sit down. And the more I walk, the more severe the pain is when I sit down [Mobility and ADL]… 'It's not the ideal situation, because it makes me little bit home bound and I cannot travel as freely as I would like to, long distance and things like that [Participation]… this summer I had high hopes of doing a lot of other things, to improve my back and things like that, but with this balloon sticking from my stomach I can't even go swimming or cycling…. Cycling might be a bit too strenuous. Although I have been told that people do go swimming with this type of tube, but I don't want to take that chance' [Psychological impact] (02, T3) | 3 Discordant |
| (8) ⇑Pain ⇓ Mobility ⇔ Participation ⇑ Psychological impact before T3 spinal fusion | 15 | 'I got no pain at all now… I am pain free in my back [Pain]… I got a walking frame that I can with difficulty walk around the house [Mobility and ADL]. I say 'with difficulty' because at times my right leg just tend to buckle slightly, because the right leg is by far the worst one… it was by far the worst affected it does tend to buckle… and so either my wife or the carer when they come, they escort me, it is they don't have to do anything, they are there as a safety precaution [Mobility and ADL]…. All in all, it's been a fairly traumatic sort of time… I keep myself in good spirits… well its happened what is the point of being miserable about it, I keep myself in good spirits and carry on [Psychological impact]… Not doing any physical activity, obviously my body is not getting tired…. Obviously at the moment, is the rest and recuperation time…' [Participation] (15, T3) | 3 Discordant |
| (9) Pain predominantly from other MSK problem. ⇓ Mobility, Psychological impact, Participation (or not mentioned) | 04, 14, 39 | 'At the moment I think I'm at a bad stage, because I've got so many things going on…. I've got the back pain. I've got the neck pain. I've got the knee, which comes out of joint. So it's not a pretty sight really [Pain]…. I'm used to going everywhere and very, sort of, doing exercises and yoga classes and everything else. And it's getting worse, because it's stopping, it's stopping, everything is going wrong…. So I've got all these things going at once [Mobility and ADL and Participation]. Which has dragged me down a bit, there's no doubt about it, it's made me quite depressed' [Psychological impact] (14, T3) | 4 Transient change |

Continued

**Table 5** Continued

| Co-Present trajectories of biopsychosocial domains | Participant ID | Participants' quotes illustrating interaction between co-present trajectories | Trajectory category (see table 2) |
|---|---|---|---|
| (10) Pain predominantly from other MSK problem. ⇓Mobility, Participation (or not mentioned) ⇑Psychological impact | 32 | 'Terrible [**Pain**] at the moment… Such a lot of pain in my knee then I have got pain in my right buttock which is more like… it is the sciatic nerve is trapped and I can't sit for longer I have to keep putting the pillow under or keep lying down… Lying down flat makes my back worse… its [Hip pain] been on and off a couple of years… some days it's [Back pain] been worse, I can't get washed and dressed… It's pain everywhere… there is nothing you can do till you get repaired [**Psychological impact**]… I can't do any housework or anything because I can't stand without hanging on to my trolley… Next week I am starting meals on wheels… It is non-existent really [General lifestyle] I could do a few things but not now… it is walking [reason] I wouldn't want to go far because it gets too painful, I walk to the car' [**Mobility and ADL and Participation**] (32, T3) | 5 No change |

ADL, activities of daily living; MSK, musculoskeletal.

except participant 38 who was in trajectory group 3 as they had transient decline before improvement in two domains. This participant gained mobility and social and recreational participation through learning to effectively use a walking aid.

In the group with co-presence type 3, where there was no improvement in pain but improvement in all other domains, participant 43 used pain relief for the persisting pain enabling other domains to improve even though he was not keen on taking painkillers, and 24 experienced pain only at night. Both were in trajectory groups 1 or 2.

In type 4 (n=5), all domains were worsening. This was the second most common type of co-presentation and mirror image of that observed in type 1. Worsening in one domain had a negative impact on the other domains. All participants were in trajectory group 5 (no change in at least two domains) except participants 33 and 41. The former experienced transient improvement in two domains but then worsened, so was in trajectory group 4. The latter was in trajectory group 3 where there was discordance. This participant had worsening pain and mobility which she had found depressing but by T3 she saw the pain, although severe, as manageable. The level of mobility required for her to maintain most of her recreational and social interests was relatively limited and she could manage this through planning, but she had difficulty with activities involving walking in groups.

Type 5 (n=2) was similar to type 4 and both participants were in trajectory group 5, but there was improvement in psychological impact. Participant 53 had specifically addressed his depression that had resulted from the NC symptoms and 06 had adapted to the pain. However, in the timeframe of the study the reported improved psychological impact did not influence the other domains.

Participant 34 with co-presence type 6 where pain did not improve and mobility deteriorated (trajectory group 5) similarly maintained positive psychological outlook but was also able to maintain social and recreational engagement despite the pain and loss of mobility.

Participant 02 (co-presence type 7) and 15 (co-presence type 8) were both in trajectory group 3 where there was discordance. The former remained envious of those who could walk faster than he could which impacted negatively on his psychological well-being and would have liked to be able undertake long distance travel which was limited by his pain. Participant 15 had spinal surgery between T2 and T3 which raised his hopes of improvement although had not yet experienced improvement.

Co-presence types 9 and 10 were similar to types 4 and 5, respectively, but the pain was not from NC but from other musculoskeletal problems.

We were particularly interested in those who still improved in social and recreational participation without improvement in all other domains. This was experienced by four participants for a variety of reasons highlighting those ongoing impairments across other domains does not necessarily impede improvement in participation. For 24 the pain was only at night, for 43 the pain could be

controlled, and for 34 and 41 it was their positive psychological outlook.

### Turning points

Thirteen participants did not mention a major change in their health. Twelve described major change in musculoskeletal problems: four described notable improvement or worsening of their pain and mobility, one had a fall that impacted well-being, four described worsening of joint problems (hip, knee and neck) and four had back surgery during the data collection year. Four participants developed other major health issues (emergency abdominal surgery, Addison's disease, chronic obstructive airways disease, insertion external feeding tube). From the data for these latter participants, we were unable to detect any interaction between their trajectory and these health events.

### Totality of change

There were many similarities between participants from across the different trajectories. The majority of participants lived with a partner with a total of 9 living alone. The majority of participants (from all categories) had a positive outlook on ageing and wanted to be active, '…
*Age is just a number, for me'* (08, T1, 70 years). A few either had an acceptance of ageing or had negative attitudes, *'You'd think I was 95 the way I'm talking to you'* (11, T2, 70 years). Most participants (from all categories) were using some form of mobility aids ranging from walking stick, wheeled walker and rollator, Nordic poles, Zimmer frame and mobility scooters. A very few (all categories) mentioned they did not want to become dependent on mobility aids or look older, *'I don't like them [laughs]. Vanity. I just suppose… It looks like you are an old man you know, or a disabled person'* (02, T1). In all categories, good weather/summer season enabled walking and outdoor activities and many reported limited activity during winter due to difficulties such as lack of confidence in walking when wet/icy and difficulty finding places to sit down. All participants reported a variety of self-made, convenient strategies for coping with their NC in at least one of their interviews. Participants predominantly used active, problem-focused strategies rather than passive ones such as avoiding painful activities. For example, they used solution-based strategies such as sitting/resting between walks or household chores, driving to local shops instead of walking, leaning on a shopping trolley to walk around, using a light-weight vacuum cleaner, and accepting help from family members with shopping and gardening. Participants who regularly (mostly daily, one 2–3 times a week) engaged in exercises specifically for their NC were found in all trajectory groups.

## DISCUSSION

For each person living with NC, we were able to identify the pattern of change at each of the three time points for each of the biopsychosocial domains of pain, mobility,

psychological impact and social participation. By looking across these patterns, we were able to identify trajectories over time for each domain. Participants in all trajectory categories had many attributes in common including attitudes to ageing and in seeking strategies to enable themselves to function optimally.

Although there were participants where improvement occurred across all domains or there was no change across all domains, many participants reported discordance with change going in different directions in different domains, or there was no change in some domains alongside change in other domains. Published quantitative evidence has demonstrated within domain discordance: for mobility, discordance between walking capacity and actual walking.[32][33] With our qualitative data, we were able to illustrate interaction between domains, for example, pain affecting mobility, with negative psychological impact and social withdrawal and vice versa. This negative cycle might explain the within domain discordance for mobility. However, we were also able to illustrate how, for example, improvement in psychological impact or social and recreational interaction happened despite pain or mobility problems.

For 10 out of 14 participants reporting improvement in social and recreational participation, this occurred alongside improvement in all other domains. This is the best scenario and the aim of interventions for people with NC. Remaining participants reported dealing in some way specifically with their pain, mobility or psychological impact which still enabled them to improve social and recreational participation without improvement in all domains. This suggests that it is possible to identify strategies to increase participation even if other domains are resistant to change. This is particularly important when considering the impact of pain. Pain is particularly problematic for this patient group and effective treatments for reducing the pain associated with NC are limited.[34] For the participants where improvement in social and recreational participation was not achieved when it might be expected, the reasons were as follows: very high expectations of what for them counted as participation; did not want to cause a problem for others; there was no loss of engagement to be regained. The other group warranting further consideration are those reporting worsening across all the domains and what can be done to break this perpetuating negative cycle.

### Categorising and understanding change over time

Our study extends the limited current longitudinal qualitative evidence on living with back pain. We found a greater variety of trajectories than in a published study with just eight participants (see background)[12] as, in contrast to this study, we found examples of change in one domain despite no change in others as well as examples of reinforcement of change across domains. The earlier study collected data at longer time intervals (12 months over 2 years). This may have resulted in less detail about change between data points than in our study resulting

in the perception of smoother trajectories. In our past qualitative study[22] which included 15 participants living with chronic back pain (excluding NC) interviewed three times in 2 years, we developed the following categories of participant's current experience of their back pain: past reminders (similar to fear avoidant behaviour), stuck and struggling, becalmed and submerged. These categories describe the overall state of the participants rather than interactions that resulted in these states. Of the 15 participants, 12 changed category during the 2 years. Most of the participants started as stuck and struggling and moved to past reminders or becalmed, trajectories similar to our current study participants in categories 1 and 2. Two participants remained stuck and struggling/ submerged, a similar trajectory to the participants at the bottom of our table 2 (trajectories of no change/worsening). Our past study included cross-sectional data only from people living with diabetes[22] who were categorised in a similar way to those with back pain.

Trajectories have been identified in qualitative studies of people living with long-term conditions other than back pain where longitudinal data has been collected. In a study of 22 participants, data were collected 15 months post-stroke and over the subsequent 4 months. Four different recovery trajectories were evident: (a) meaningful recovery, (b) cycles of recovery and decline, (c) ongoing disruption and (d) gradual, ongoing decline.[35] The first (a) is similar to our progressive improvement trajectory, (b) may be similar to our discordant trajectories, and (c) and (d) have some similarity to our category of no change; the difference may be due to the natural history of the NC where NC only worsens for a few.[6] However, trajectories identified depend on the focus of analysis. A longitudinal interview study with 21 participants diagnosed with a chronic illness (eg, diabetes, rheumatoid arthritis) focused on illness self-management.[36] It was found that participants go through the following phases: seeking effective self-management strategies, considering costs and benefits, creating routines and plans of action, and negotiating self-management that fits one's life. In our study, we found all these strategies being used but our analysis focus was not self-management.

### Categorising to predict outcome and complex interaction

We found many similarities between the participants in the five categories of trajectories and a lack of features that could be used to identify members of a particular trajectory which could be used to predict outcome. Many studies on back pain have analysed quantitative data using methods for categorising such as latent class analysis, seeking categories of patients that would allow matching of interventions to patient groups.[37–39] Study participants can be categorised into clinically meaningful groups and category membership may correlate with outcome.[40] However, interaction between category membership and intervention is difficult to demonstrate.[38] A longitudinal study of spinal stenosis found factors predictive of subjective improvement, but none were modifiable.[41]

If improved social and recreational participation is our goal, our study would suggest that this is likely to be co-present with and interact with improvement across the domains of pain, mobility and psychological impact and not improve when there is no co-presence of improvement across the other domains. This interaction between domains is like that described in a cross-sectional interview study with people living with NC.[5] Although our data suggest that for some participants improvement in one domain leads to improvement in another domain, this interaction is in the timeframe of hour-to-hour or day-to-day or week-to-week as occurs in complex systems.[42 43] This, what we have called co-presence, is different from the prediction of outcome over many months or years that is sought in the studies described earlier. However, our study also provides examples of participants where one factor did prevent or enable improvement, but this was a relatively small number of participants and the attributes varied. We suggest that seeking individual level factors that predict of outcome for people living with NC is unlikely to be helpful. However, we need to remain alert to external determinants of outcome. For example, in a US study, type of health insurance influenced rate of improvement for people with low back pain.[44]

### Strengths and limitations

We collected a relatively large qualitative data set with higher-than-expected participation rates in later interviews. This may have been because the repeat encounters generated a relationship of trust between researcher and participant.[45] Although there are differences when interviews are conducted by telephone rather than face-to-face,[46] our telephone interviews provided rich data for our analysis, perhaps helped by the first interview being face-to-face.

At each interview, we focused on how the participant was at the time of the interview. However, in interviews people naturally tell stories and these stories include accounts of change that has occurred.[47] What was current was sometimes difficult to disentangle and we may have interpreted later data differently given our understanding of the participant from earlier interviews.[23] The data we included in the current analysis were from participants providing three interviews of rich data so they may differ from those in our study providing less data. As we proceeded through data analysis, we chose our path based on our interest in the complex nature of health, the lived experience of conditions such as NC and successful ageing. Alternative data analysis paths through this data set are possible. Analysis could be tested further by interrogating the remaining unanalysed data. Reduction of data to mini-statements loses richness and nuances in data, but we retained the ability to go back directly to the framework and the data for fuller summaries and illustrative quotes. Data reduction enabled comparison across the whole data set.

Although a longitudinal study, follow-up was only for 12 months, a relatively short time. NC symptoms change

over time. Some of our participants had already started to experience improvement in their symptoms at the start of the study. This is in line with findings of a cohort study of people with moderate NC.[6]

Participants could be categorised in different ways and there was overlap between categories. This is a common finding when categorising people in both qualitative studies[22] and quantitative studies.[37]

## Implications for clinical practice

There may be benefit in those providing therapy for patients with NC to identify the trajectory of their patient in order to tailor treatment. This requires careful listening as changes may be quite small so hard to discern. Although it is useful to categorise patients as we have done in this study, these categories are just useful labels to think with and guide discussions with patients. As therapists know, patient presentation can often seem messy. Having the categories of trajectory and how they potentially interact in mind may help tease out what is going on for people and identify where there is opportunity to reinforce positive change or modulate deterioration, but we need to beware of shoehorning people into categories as we could miss important changes in trajectory in one or more domains. Our study suggests therapists, while following clinical guidance for patients with NC, also need to individually tailor their interventions for patients to optimise participation in social and recreation activities that are fundamental to healthy ageing.

**Contributors** SEL and EW are health service researchers and physiotherapists and led the BOOST programme of work including this study and the BOOST trial. FG is a general medical practitioner, medical sociologist and qualitative researcher and led the qualitative study team that included LW, a qualitative health researcher; CS, a physiotherapist and a qualitative researcher; and GB, a physiotherapy researcher. All contributed to the qualitative study design, reading data, thematic and comparative analysis, synthesis, interpretation and reporting. LW undertook all interviews. LW and FG reviewed the interview schedules to improve the clarity and flow of the questions. CS worked with FG on analysis with GB providing independent review of coding/summaries. SEL is the chief investigator, senior author and guarantor. Any discrepancies were resolved through discussion. EW and SEL reviewed each step of the analysis and provided feedback to FG, LW, CS and GB.

**Funding** This research is funded by the NIHR Programme Grants for Applied Research (reference: PTC-RP-PG-0213-20002). Preparatory work for the programme of research was supported by the Collaboration for Leadership in Applied Health Research and Care Oxford at Oxford Health NHS Foundation Trust. SEL and EW received funding from the Collaboration for Leadership in Applied Health Research and Care Oxford at Oxford Health NHS Foundation Trust and are supported by the NIHR Biomedical Research Centre, Oxford.

**Competing interests** None declared.

**Patient and public involvement** Patients and/or the public were involved in the design, or conduct, or reporting, or dissemination plans of this research. Refer to the Methods section for further details.

**Patient consent for publication** Obtained.

**Ethics approval** This study involves human participants and ethics approval was given by the London-Brent National Research Ethics Committee (REC number 16/LO/0349) on 3 March 2016. Participants gave informed consent to participate in the study before taking part.

**Provenance and peer review** Not commissioned; externally peer reviewed.

**Data availability statement** All data relevant to the study are included in the article or uploaded as supplementary information.

**ORCID iDs**
Frances Griffiths http://orcid.org/0000-0002-4173-1438
Cynthia Srikesavan http://orcid.org/0000-0002-3540-8052
Esther Williamson http://orcid.org/0000-0003-0638-0406

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
