## [Reviewer comments · BMJ Open]

ARTICLE DETAILS

TITLE (PROVISIONAL)	Longitudinal qualitative study of living with neurogenic claudication
AUTHORS	Griffiths, Frances; Srikesavan, Cynthia; Ward, Lesley; Boniface, Graham; Williamson, Esther; Lamb, Sarah

VERSION 1 – REVIEW

REVIEWER	Ammendolia, Carlo Mount Sinai Hospital; University of Toronto, Canada, Medicine
REVIEW RETURNED	08-Jan-2022

GENERAL COMMENTS	Thank you for the opportunity to review this comprehensive qualitative investigation on biopsychosocial trajectories among people living with neurogenic claudication General Comments Overall, the paper is very well written although many of the tables were difficult to read and interpret (see specifics below). Although this study did not shed a lot of new light on this topic, the findings did emphasize the high variability in change over time in the biopsychosocial domains/outcomes in this population. Perhaps a deeper dive into patients' attitudes and beliefs and expectations (of treatment success and future health status) at baseline may have been more fruitful in explaining some of the observed/described trajectories. Finally, more thought on possible theoretical explanations of the findings in the discussion would enrich the paper. Specific comments/suggestions 1. The domain "Pain". In the background section the authors correctly describe the variability of symptoms in neurogenic claudication. Although many describe their symptoms as pain, a significant number describe their symptoms as lower extremity numbness, tingling, pins and needles, fatigue, heaviness and not pain. I am assuming that these other symptoms were also considered in this domain and therefore it may be more accurate to name this domain as "symptoms".2. The selection of the 30 participants. Please provide more information on how the final selection was made since your conclusions are based on this sample. Would selecting a different sample among the 49 patients with completed datasets resulted in the same findings/conclusions? Other qualitative studies use saturation of information to justify sample size but in this study, it is not clear. The authors do provide a reference to justify their
--

	sample size (Malterud 2016), but I think it is important enough that a justification be provided in the manuscript. 3. Table 1. Participant characteristics. The inclusion of biopsychosocial variables in this table would provide more meaningful information about the study sample and can help to better contextualize the described trajectories. 4. Table 2. Participant trajectories in the biopsychosocial domains. This table is very difficult to interpret without the legend right along side the table on same page. I would suggest adding text in the colour boxes to describe the change. 5. All supplemental tables should have titles. Two of the tables (without titles) are extremely difficult to read due to very small font sizes. 6. The impact of treatment received. There is no mention of the potential impact of the treatment received on the trajectories. All participants received physiotherapy, about half received the Boost 12-week program and other half 1-3 physiotherapy sessions. At first glance the participants outcomes appear no better than that described in natural history studies. The clinical practice guidelines cited, Bussi�eres et al suggests that non operative interventions can impact functional outcomes. 7. Baseline attitudes and beliefs about their condition and their treatment expectations. This information appeared to be collected at baseline. In other studies (both surgical and nonsurgical) these factors appeared strongly predictive of outcomes in this population, often more so than interventions received (Ammendolia 2017). Was there any sense from the interviews these factors played a role in the trajectories and if not, more discussion is warranted given the findings in other studies of neurogenic claudication and chronic LBP. 8. Discussion and study conclusions. The authors provide a good summary of the study findings but omit an overarching conclusion. Suggest using the conclusion in the abstract. From the data presented, the main take home message is that the trajectories are highly variable in this population with interaction and change between domains often discordant. I suggest less detail on summarizing the findings and more thought on why there is high variability and discordance in this population? What are some of the possible theoretical explanations? There are other studies (Conway 2011, Schneider 2019, Smuck 2018) that also have demonstrated discordance where despite improvements in pain and physical capacity this did not translate in improvements in physical performance. This could be due to modified routines and lifestyle that become established after being inflicted with NC that are then resistant to change? These findings and those found in this paper speak to the underlying complex interaction of physical, functional, and psychosocial factors in this population. Some possible explanatory pathways would enrich the paper. 9. Study limitations and implications for clinical practice. Well done and nothing to add.
--	---

REVIEWER	Baca, Kira Palmer College of Chiropractic, Research
REVIEW RETURNED	28-Jan-2022

GENERAL COMMENTS

Thank you for the opportunity to peer review this submitted manuscript describing changes in participant experiences living with NC though time. I wish to thank and congratulate the authors on conducting a study in an area where more research is clearly needed. Below you will find comments designed to help improve the submission. my review, for the submitted manuscript by Frances Griffiths, et al.

General Comments:

1. Several grammatical errors and word misuse were found, and several notes below. Please carefully review the entire manuscript for such errors.
2. No conclusion was present in this submission. Please consider including a summary of the main findings and potential clinical application in the abstract.

Background:

1. General
 - a. Please provide context for the sentence: "The impact of back and leg pain consistent with NC on health status is more marked than for people with other patterns of back and leg pain" Currently, this statement is far too broad. In what ways does NC have a more marked effect on people (pain severity, disability, quality of life, functional loss, influence on comorbidity, etc.) than those suffering from other patterns of back and leg pain?
 - b. The reference used here: "Among 146 people with NC followed up for 3 years..." may be too specific for use in the background, and benefit from deeper dissection if used and explored in the discussion section instead.
 - c. Asking whether "symptoms may lead to older people being unable to age successfully" in this section of the background should be removed. It is redundant with the final sentence in the 2nd paragraph and aging successfully hasn't been defined. It suggests a study purpose and raises unnecessary questions with no rationale for why the authors conducted the study and no definition of aging successfully by this point in the manuscript.
 - d. The last sentence of the second paragraph in the first section of the background is awkwardly worded. Please revise for clarity.
2. Qualitative evidence on living with back pain
 - a. Consider rewording the second sentence in this section for clarity.
 - b. Last part of the last sentence of the first paragraph in this section needs to be revised for clarity,

namely “and those [who] experienced pain relief”.
“Use” was replaced with “who”.

Methods:

1. Study Design
 - a. Please report locations for T2 interviews.
2. Participants and Recruitment
 - a. It would be beneficial to your readers to have the context of inclusion/exclusion criteria used by the BOOST trials explained here.
 - b. Please inform the reader whether the 1:1 ratio for treatment allocation was confirmed for this study.
3. Data collection
 - a. In this section of the submission, T1 interviews are referred to for the first time as Time 1 interviews. Please consider adjusting the introduction of T1-3 interviews as Time 1-3 interviews, and T1-3 thereafter.
 - b. Please add details pertaining to how the interviews were refined for individual participants after their preliminary T1 interview is needed.
 - c. Please indicate how confidentiality was maintained regarding the audio-recordings used in this study
4. Data Analysis
 - a. In the first sentence of this section, stating that this study “constantly” compared data is vague. Is there an amount of time to report here? Otherwise, consider removing the descriptor, as it leads the reader to question how much or often.
 - b. It is unclear what the sentence that starts with “This approach puts to one side...” in this section is trying to convey. Does this mean that the reinterpretation of earlier events is not considered in the study? Please consider rewording for clarity.
 - c. Step 1
 - i. Of the 30 complete interview sets, the study selected 10 T2 and 10 T3 interviews to transcribe. Please report the process for how those 10 T2 and T3 interviews were selected from the original 30 interview sets?
 - ii. Please briefly explain what a pen portrait is for readers who are not familiar with this process.
 - d. Step 4
 - i. An example provided in the section listed as “Totality of change” would be helpful context to clarify your description to readers.
 - ii. If the words “looked at” are consistent with the term “identified” (or another more clear descriptor) I strongly suggest using it, to be more consistent with scientific writing practices.

Results:

1. Participant Characteristics
2. Table 1
 - a. The table includes the value, 59 total participants under Interview cohort heading. In the text, this number was reported as 60. Please address this discrepancy.
3. Table 2
 - a. In the table and legend, the two light green shades (“No change” and “Improvement remained same”) are too similar to be visually effective. Consider using a different color or making the shades less similar.
 - b. In the legend, mention is made to a metric you decided not to measure (sleep). Consider removing sleep from the descriptions in the legend.
 - c. On the legend, the visual aid in the righthand column is a great idea. Well done.
4. Analysis Results
 - a. *Analysis Steps 1-3:*
 - i. Please expand on the sentence “Using this categorisation some participants could be in more than one category...”. For example, how was a final decision on which category these participants were placed made? Please elaborate and explain.
 - b. *Analysis Step 4:*
 - i. In the sentence “In the group with co-presence type 3, where there was no improvement in pain but improvement in all other domains, participant 43 used pain relief for the persisting pain enabling other domains to improve and 24 experienced pain only at night. Both were in trajectory groups 1 or 2.” Is the “pain relief” used by participant 43 referring to a prescription or over the counter medication? Please specify.

Discussion:

1. *Categorising and understanding change over time*
 - a. “This may have resulted in less detail about change between data points [that] in our study resulting in the perception of a smoother trajectories.” In this sentence, “that” should be replaced with “than”.
 - b. The submission discusses the similarities between this study’s trajectories, and categories (a) and (c) of a cited study (31). Please expand on why this study has drawn such connections between these categories, and this study’s trajectories. Specifically, how does this study justify their

	trajectory with no improvement or worsening condition across all domains to not be best suited to category (d) (“gradual, ongoing decline”).
--	--

VERSION 1 – AUTHOR RESPONSE

Reviewer: 1

Dr. Carlo Ammendolia, Mount Sinai Hospital; University of Toronto, Canada Comments to the Author:

Thank you for the opportunity to review this comprehensive qualitative investigation on biopsychosocial trajectories among people living with neurogenic claudication

General Comments

- A. Overall, the paper is very well written although many of the tables were difficult to read and interpret (see specifics below). Although this study did not shed a lot of new light on this topic, the findings did emphasize the high variability in change over time in the biopsychosocial domains/outcomes in this population. Perhaps a deeper dive into patients’ attitudes and beliefs and expectations (of treatment success and future health status) at baseline may have been more fruitful in explaining some of the observed/described trajectories.

Response: Thank you for raising the issue of attitudes, beliefs, and expectations. Please see response to 7 below, about attitudes and beliefs.

To explore expectations of treatment, we looked again at the desired treatment outcomes of participants. We noted that approximately half had an absolute desired outcome (e.g. To be able to do long walks (up to 3 miles) that she enjoys without the need to take breaks and sit (participant 47); Resume physical and social activities as before (participant 3)) Other participants had relative desired outcomes (e.g. Get better sleep (participant 4); to improve her leg strength (participant 41)). We looked to see whether these different types of outcomes clustered in one part of Table 2 – the trajectories. We found each type was evenly spread across the different trajectories. We therefore have not pursued this further.

- B. Finally, more thought on possible theoretical explanations of the findings in the discussion would enrich the paper.

Response: Thank you for this prompt to review our theoretical explanation. For this condition specific, empirical study, we found it difficult to decide on the balance between drawing out new learning from the empirical data and exploring theoretical explanation. In the discussion, we have added some clarifications in the section on *Categorising to predict outcome*, to draw attention more clearly to complexity as theoretical explanation. The section title now reads:

Categorising to predict outcome and complex interaction

The edited part of the section now reads:

If improved social and recreational participation is our goal, our study would suggest that this is likely to be co-present with and interact with improvement across the domains of pain, mobility and psychological impact and not improve when there is no co-presence of improvement across the other domains. This interaction between domains is like that described in a cross-sectional interview study with people living with NC (5). Although our data suggests that for some participants improvement in one domain leads to improvement in another domain, this interaction is in the timeframe of hour-to-hour or day-to-day or week-to-week as occurs in complex systems (38, 39). This, what we have called co-presence, is different from the prediction of outcome over many months or years that is sought in the studies described above.

Specific comments/suggestions

1. The domain "Pain". In the background section the authors correctly describe the variability of symptoms in neurogenic claudication. Although many describe their symptoms as pain, a significant number describe their symptoms as lower extremity numbness, tingling, pins and needles, fatigue, heaviness and not pain. I am assuming that these other symptoms were also considered in this domain and therefore it may be more accurate to name this domain as "symptoms".

Response: We have clarified that this domain is pain. The text reads:

Participants reported a range of symptoms including numbness, tingling and heaviness. However, the symptom that consistently predominated across all participant's accounts of their experience of NC was pain; we therefore summarised the experience of pain in the framework.

2. The selection of the 30 participants. Please provide more information on how the final selection was made since your conclusions are based on this sample. Would selecting a different sample among the 49 patients with completed datasets resulted in the same findings/conclusions? Other qualitative studies use saturation of information to justify sample size but in this study, it is not clear. The authors do provide a reference to justify their sample size (Malterud 2016), but I think it is important enough that a justification be provided in the manuscript.

Response: we have added further detail to Step 1 of Data analysis:

We initially selected 30 complete interview sets which provided rich data and after initial analysis (see Step 2 below) reviewed whether later interview sets were revealing new themes; we determined we had achieved data saturation (reference: Guest G, Bunce A, Johnson L. How Many Interviews Are Enough?: An Experiment with Data Saturation and Variability *Field Methods*. 2006;18(59)).

3. Table 1. Participant characteristics. The inclusion of biopsychosocial variables in this table would provide more meaningful information about the study sample and can help to better contextualize the described trajectories.

Response: we have added to Table 1 measures related to NC to provide this context and refer to the table in the text on Participant characteristics.

4. Table 2. Participant trajectories in the biopsychosocial domains. This table is very difficult to interpret without the legend right along side the table on same page. I would suggest adding text in the colour boxes to describe the change.

Response: We provide Table 2 in two formats: with the legend alongside and with the legend underneath. We leave it to the editor to decide which will fit best in the journal lay out.

5. All supplemental tables should have titles. Two of the tables (without titles) are extremely difficult to read due to very small font sizes.

Response: We have added titles to each of these tables. The font sizes are small to manage the size of each table; readers can zoom in to read the text in the current format. We would appreciate further editorial guidance on this.

6. The impact of treatment received. There is no mention of the potential impact of the treatment received on the trajectories. All participants received physiotherapy, about half received the Boost 12-week program and other half 1-3 physiotherapy sessions. At first glance the participants outcomes appear no better than that described in natural history studies. The clinical practice guidelines cited, Bussi eres et al suggests that non operative interventions can impact functional outcomes.

Response: In an earlier draft of this paper, our Table 1 included whether the participant was in the intervention or control intervention arm of the clinical trial. Examining it by eye it demonstrated that there was no pattern connecting participant allocation and trajectory. We decided to omit this as we felt it distracted the reader from the qualitative nature of the study where we are seeking evidence of patient experience rather than specifically effect of interventions.

7. Baseline attitudes and beliefs about their condition and their treatment expectations. This information appeared to be collected at baseline. In other studies (both surgical and nonsurgical) these factors appeared strongly predictive of outcomes in this population, often more so than interventions received (Ammendolia 2017). Was there any sense from the interviews these factors played a role in the trajectories and if not, more discussion is warranted given the findings in other studies of neurogenic claudication and chronic LBP.

Response: Attitudes and beliefs about the participants' condition was not part of the interview schedule; we were interested in the experience of living with NC and how this experience with NC changed with time. We did ask about treatment expectations (see response to A above).

We have carefully reread Ammendolia 2017. This is an excellent paper that explores the experience of NC and its treatments. As this paper would not have been included in the meta-ethnography referred to in the second section of the introduction we have included more details about this.

Our response to B (see above) includes the addition to the discussion of a sentence about Ammendolia 2017 and the issue of interaction of domains.

8. Discussion and study conclusions. The authors provide a good summary of the study findings but omit an overarching conclusion. Suggest using the conclusion in the abstract.

Response: We have checked the authors instructions for BMJ Open, and they do not suggest including an overarching conclusion at the end of the discussion.

9. From the data presented, the main take home message is that the trajectories are highly variable in this population with interaction and change between domains often discordant. I suggest less detail on summarizing the findings and more thought on why there is high variability and discordance in this population? What are some of the possible theoretical explanations? There are other studies (Conway 2011, Schneider 2019, Smuck 2018) that also have demonstrated discordance where despite improvements in pain and physical capacity this did not translate in improvements in physical performance. This could be due to modified routines and lifestyle that become established after being inflicted with NC that are then resistant to change? These findings and those found in this paper speak to the underlying complex interaction of physical, functional, and psychosocial factors in this population. Some possible explanatory pathways would enrich the paper.

Response: Thank you for raising this issue and for your helpful references. In the second paragraph of the discussion, we have clarified how our results contributes to understanding the discordance you describe. It now reads:

Although there were participants where improvement occurred across all domains or there was no change across all domains, many participants reported discordance with change going in different directions in different domains, or there was no change in some domains alongside change in other domains. Published quantitative evidence has demonstrated within domain discordance: for mobility, discordance between walking capacity and actual walking (reference: Conway J, Tomkins CC, Haig AJ. Walking assessment in people with lumbar spinal stenosis: capacity, performance, and self-report measures. *Spine J.* 2011;11(9):816-823. doi:10.1016/j.spinee.2010.10.019. Smuck M, Muaremi A, Zheng P, et al. Objective measurement of function following lumbar spinal stenosis decompression reveals improved functional capacity with stagnant real-life physical activity. *Spine J.* 2018;18(1):15-21. doi:10.1016/j.spinee.2017.08.262). With our qualitative data, we were able to illustrate interaction between domains, for example, pain affecting mobility, with negative psychological impact and social withdrawal and vice versa. This negative cycle might explain the within domain discordance for mobility. However, we were also able to illustrate how, for example, improvement in psychological impact or social and recreational interaction happened despite pain or mobility problems.

10. Study limitations and implications for clinical practice. Well done and nothing to add.

Reviewer: 2

Dr. Kira Baca, Palmer College of Chiropractic Comments to the Author:

*** Please find comments from this reviewer in the attached file *** Thank you for the opportunity to review your work.

Reviewer: 1

Competing interests of Reviewer: I have no competing interests

Reviewer: 2

Competing interests of Reviewer: None

Peer Review - bmjopen-2021-060128_Proof_hi

Thank you for the opportunity to peer review this submitted manuscript describing changes in participant experiences living with NC though time. I wish to thank and congratulate the authors on conducting a study in an area where more research is clearly needed. Below you will find comments designed to help improve the submission. my review, for the submitted manuscript by Frances Griffiths, et al.

General Comments:

1. Several grammatical errors and word misuse were found, and several notes below. Please carefully review the entire manuscript for such errors.

Response: Thank you. Our apologies for the errors. We have endeavoured to remove them.

2. No conclusion was present in this submission. Please consider including a summary of the main findings and potential clinical application in the abstract.

Response: We have checked the authors' instructions for BMJ Open, and they do not suggest including a conclusion at the end of the discussion. Our abstract conclusion has a summary of main findings and potential clinical application.

Background:

1. General

a. Please provide context for the sentence: "The impact of back and leg pain consistent with NC on health status is more marked than for people with other patterns of back and leg pain" Currently, this statement is far too broad. In what ways does NC have a more marked effect on people (pain severity, disability, quality of life, functional loss, influence on comorbidity, etc.) than those suffering from other patterns of back and leg pain?

Response: To provide context the text now reads:

The impact of back and leg pain consistent with NC on health status is more marked than for people with other patterns of back and leg pain; for older adults, compared to those who report back pain only or have leg pain which is not consistent with NC, those with NC report lower quality of life and are more likely to be frail, be less confident to walk ½ a mile and to report a decline in their mobility over the last year (1).

b. The reference used here: "Among 146 people with NC followed up for 3 years..." may be too specific for use in the background, and benefit from deeper dissection if used and explored in the discussion section instead.

Response: We have rephrased this sentence, so it conveys to the reader the variation in trajectory that is already evidenced. This is important as our paper provides qualitative detail about trajectories. The sentence now reads:

The variation in symptom trajectory for people with NC is illustrated by a study of 146 people with NC followed up for 3 years (without surgical treatment), symptoms improved for about one third of patients, remained unchanged for half the patients and worsened in a few (6).

c. Asking whether “symptoms may lead to older people being unable to age successfully” in this section of the background should be removed. It is redundant with the final sentence in the 2nd paragraph and aging successfully hasn't been defined. It suggests a study purpose and raises unnecessary questions with no rationale for why the authors conducted the study and no definition of aging successfully by this point in the manuscript.

Response: Thank you for pointing this out. We have removed the sentence from the first paragraph. We have edited the latter part of the third paragraph to include this sentence. The third paragraph of now reads:

Qualitative evidence on living with back pain

The interaction physical and mental health and social and recreational engagement has been explored in qualitative studies of people living with low back pain and synthesised in a meta-ethnography (11). The interrelated themes identified were: impact of pain on self, use of coping strategies and impact of pain on engagement with others. The meta-ethnography concluded that chronic low back pain is complex, dynamic and multidimensional with those living with the pain experiencing distressing pain and loss with lowered self-worth, stigma, depression, premature aging, and fear of the future. A later cross-sectional qualitative study of people with NC, suggests that pain, physical ability and emotions are inter-related and at times inseparable (5). The meta-ethnography identified a lack of longitudinal studies. A subsequent longitudinal study interviewed eight people living with back pain annually over two years (12). Although pain and loss impacted on all areas of the participants lives, two different patterns of change over time were identified: participants who saw their pain a physical problem and a cause of loss and did not re-establish social and recreational activities, and those who experienced pain relief and were less enmeshed in the pain experience so able to look to the future (12).

d. The last sentence of the second paragraph in the first section of the background is awkwardly worded. Please revise for clarity.

Response: This is now revised along with the edits for 1c above.

2. Qualitative evidence on living with back pain

a. Consider rewording the second sentence in this section for clarity.

b. Last part of the last sentence of the first paragraph in this section needs to be revised for clarity, namely “and those [who] experienced pain relief”. “Use” was replaced with “who”.

Response to 2 a and 2 b: This is now revised along with the edits for 1c above.

Methods:

1. Study Design

a. Please report locations for T2 interviews.

Response: we have added that T² interviews were by telephone.

2. Participants and Recruitment

a. It would be beneficial to your readers to have the context of inclusion/exclusion criteria used by the BOOST trials explained here.

Response: Thank you for raising this. We have added information about the inclusion/exclusion criteria used by the BOOST trial in the section that describes the trial. The text reads:

Participants aged 65 years or over with the clinical presentation of NC symptoms were recruited from primary care practices of the National Health Service (NHS) in the United Kingdom (UK). People living in residential care or nursing homes, those registered blind and those with a terminal health condition or on a waiting list for surgery or unable to walk 3m without the support of another person or unable to provide informed signed consent were not included.

b. Please inform the reader whether the 1:1 ratio for treatment allocation was confirmed for this study.

Response: Thank you for drawing our attention to checking the relevant text and table. There were errors in Table 1 which we have now corrected. We have added the following to the description of the BOOST trial:

Participants were randomised in a 2:1 ratio (BOOST programme: BPA).

3. Data collection

a. In this section of the submission, T1 interviews are referred to for the first time as Time 1 interviews. Please consider adjusting the introduction of T1-3 interviews as Time 1-3 interviews, and T1-3 thereafter.

Response: We have corrected this.

b. Please add details pertaining to how the interviews were refined for individual participants after their preliminary T1 interview is needed.

Response: We have clarified and given an example:

Before T2 and T3 interviews we reviewed for each participant the data captured in their earlier interviews and made notes about the participant and their context. We used this knowledge in the interviews to rapidly establish rapport on the telephone and enable quick clarifications. For example:

Participant: I go to my club.

Interviewer: Is that the ladies club you mentioned last time we met?

c. Please indicate how confidentiality was maintained regarding the audio-recordings used in this study

Response: We have added the following:

All interviews were audio-recorded onto encrypted recorders and stored securely at University of Oxford. During transcription/summarising all personal identifiers were removed.

4. Data Analysis

a. In the first sentence of this section, stating that this study “constantly” compared data is vague. Is there an amount of time to report here? Otherwise, consider removing the descriptor, as it leads the reader to question how much or often.

Response: We have removed the word ‘constantly’.

b. It is unclear what the sentence that starts with “This approach puts to one side...” in this section is trying to convey. Does this mean that the reinterpretation of earlier events is not considered in the study? Please consider rewording for clarity.

Response: We have edited this to clarify:

In our analysis of data at each time point we used an analysis approach we had previously developed (18) and applied (22). We focused on current experience and put to one side data where the participant was interpreting past-experience. This allowed us to compare what was current at each time point and to manage the volume of complex data (28).

c. Step 1

i. Of the 30 complete interview sets, the study selected 10 T2 and 10 T3

interviews to transcribe. Please report the process for how those 10 T2 and T3 interviews were selected from the original 30 interview sets?

Response: We have explained the process in more detail. The text now reads:

We started analysis on complete interview sets (T1, T2 and T3). We selected sets where the participant provided rich data about the experience of NC. After selecting 30 sets (90 interviews) and undertaking initial analysis (see Step 2 below) we read and discussed our remaining data to check whether it was revealing new themes; we determined we had achieved data saturation(29). All T1 interviews were transcribed verbatim along with the first 10 T2 and 10 T3 interviews we analysed. The remaining interviews were analysed directly from audio recordings.

ii. Please briefly explain what a pen portrait is for readers who are not familiar with this process.

Response: We have edited and added to the text so it now reads:

To immerse ourselves in this large volume of data we initially wrote structured pen portraits(30) at each time point for ten data sets. This involved reading and rereading the data, clarifying our focus on current experience related to NC, summarising participant experiences and context with illustrative quotes, and developing a structure that worked across the 10 data sets.

d. Step 4

i. An example provided in the section listed as “Totality of change” would be helpful context to clarify your description to readers.

Response: We have expanded this section and linked it to edits within our introduction. It now reads:

From the whole change framework we looked at factors that were not inseparable from trajectory category although are likely to be interrelated (5). Factors included whether they lived alone, how they considered aging and use of mobility aids, and strategies for coping with NC in their daily lives. We looked for what was similar and different between participants in the different trajectory categories.

ii. If the words “looked at” are consistent with the term “identified” (or another more clear descriptor) I strongly suggest using it, to be more consistent with scientific writing practices.

Response: We have replaced the use of “looked at” with a more precise term throughout the paper.

Results:

1. Participant Characteristics

2. **Table 1**

a. The table includes the value, 59 total participants under Interview cohort heading. In the text, this number was reported as 60. Please address this discrepancy.

Response: Apologies for the confusion. The error is in our methods. We have now corrected the text in the methods, so it reads:

To ensure sufficient information power (24), we aimed for 30 participants retained at 12 months and so aimed to recruit approx. 60 participants to allow for 50% attrition.

3. **Table 2**

a. In the table and legend, the two light green shades (“No change” and “Improvement remained same”) are too similar to be visually effective. Consider using a different color or making the shades less similar.

Response: We have changed the light green shade of ‘Improvement remained same’ to light purple.

b. In the legend, mention is made to a metric you decided not to measure (sleep). Consider removing sleep from the descriptions in the legend.

Response: We have removed ‘Sleep’ from the descriptions in the legend

c. On the legend, the visual aid in the righthand column is a great idea. Well done.

4. Analysis Results

a. *Analysis Steps 1-3:*

i. Please expand on the sentence “Using this categorisation some participants could be in more than one category...”. For example, how was a final decision on which category these participants were placed made? Please elaborate and explain.

Response: We have edited the Methods section for step 3 to make it clearer what we did. This now reads:

In a new table we presented for each domain the colour coded trajectory for each participant. Each team member then independently, by eye, ordered the participants so those with the most similar combination of trajectories were adjacent. The team then compared their resulting tables, discussed similarities and differences, and agreed on one version of the table. Using this table, we then independently categorised the participants into groups based on similarity of trajectories and discussed and agreed categorisation.

By clarifying the process in the methods we hope the reader has a clearer understanding of the results paragraph that starts: “When placing participants adjacent to other participants based on the similarity of trajectory,...”

We have edited the text “Using this categorisation some participants could be in more than one category...” so it now reads:

This categorisation was developed from one way of ordering the participants (see above). It was therefore expected that some participants could be in more than one category (for example participant 34 could be in category 2 or 5).

b. *Analysis Step 4:*

i. In the sentence “In the group with co-presence type 3, where there was no improvement in pain but improvement in all other domains, participant 43 used pain relief for the persisting pain enabling other domains to improve and 24 experienced pain only at night. Both were in trajectory groups 1 or 2.” Is the “pain relief” used by participant 43 referring to a prescription or over the counter medication? Please specify.

Response: On rereading Supplementary file 3 we realise we omitted a relevant detail about participant 43 – that he was not keen on using painkillers. We have added this detail to the manuscript. However, we have not added information about whether the pain relief was prescribed or over the counter does not seem relevant to the focus of our analysis. If the reader is interested to know this detail, they can read Supplementary file 3 where it mentions use of Paracetamol.

1. *Categorising and understanding change over time*

a. “This may have resulted in less detail about change between data points [that] in our study resulting in the perception of a smoother trajectories.” In this sentence, “that” should be replaced with “than”.

Response: We have corrected this error

b. The submission discusses the similarities between this study’s trajectories, and categories (a) and (c) of a cited study (31). Please expand on why this study has drawn such connections between these categories, and this study’s trajectories. Specifically, how does this study justify their trajectory with no improvement or worsening condition across all domains to not be best suited to category (d) (“gradual, ongoing decline”).

Response: We have expanded this section drawing in other longitudinal qualitative studies to explore further the similarities between trajectories (or not) for people living with long-term conditions. It now reads as follows:

Categorising and understanding change over time

Our study extends the limited current longitudinal qualitative evidence on living with back pain. We found a greater variety of trajectories than in a published study with just eight participants (see background) (12) as, in contrast to this study, we found examples of change in one domain despite no change in others as well as examples of reinforcement of change across domains. The earlier study collected data at longer time intervals (12 months over 2 years). This may have resulted in less detail about change between data points than in our study resulting in the perception of smoother trajectories. In our past qualitative study (22) which included 15 participants living with chronic back pain (excluding NC) interviewed three times in two years, we developed the following categories of participant’s current experience of their back pain: past reminders (similar to fear avoidant behaviour), stuck and struggling, becalmed, and submerged. These categories describe the overall state of the participants rather than interactions that resulted in these states. Of the 15 participants, 12 changed category during the 2 years. Most of the participants started as stuck and struggling and moved to past reminders or becalmed, trajectories similar to our current study participants in categories 1 and 2. Two participants remained stuck and struggling/submerged, a similar trajectory to the participants at the bottom of our Table 2 (trajectories of no change/worsening). Our past study included cross-sectional data only from people living with diabetes (22) who were categorised in a similar way to those with back pain.

Trajectories have been identified in qualitative studies of people living with long-term conditions other than back pain where longitudinal data has been collected. In a study of 22 participants, data was collected 15 months post stroke and over the subsequent 4 months. Four different recovery trajectories were evident: (a) meaningful recovery, (b) cycles of recovery and decline, (c) ongoing disruption, (d) gradual, ongoing decline (33). The first (a) is similar to our progressive improvement trajectory, (b) may be similar to our discordant trajectories, (c) and (d) have some similarity to our category of no change; the difference may be due to the natural history of the NC where NC only worsens for a few (6). However, trajectories identified depend on the focus of analysis. A longitudinal interview study with 21 participants diagnosed with a chronic illness (e.g., diabetes, rheumatoid arthritis) focused on illness self-management (34). It was found participants go through the following phases: seeking effective self-management strategies, considering costs and benefits, creating routines and plans of action, and negotiating self-management that fits one’s life. In our study we found all these strategies being used but our analysis focus was not self-management.

VERSION 2 – REVIEW

REVIEWER	Baca, Kira Palmer College of Chiropractic, Research
REVIEW RETURNED	03-Jun-2022

GENERAL COMMENTS	Thank you for addressing each concern and clearly articulating the changes that were made.
--

VERSION 2 – AUTHOR RESPONSE

Authors' response:

- Table 4 and Supplementary file 4 no longer include age and gender
- We have changed the text in the manuscript from “(See Supplementary file 2 for an example)” to “(Example available on request)” and we have removed the file from our submission
- After removing Supplementary file 2 and revising Table 4, we have updated the main document. The submission now contains 5 Tables and 3 Supplementary files
- We have updated the research checklist